# First Report of Blood Fluke Pathogens with Potential Risk for Emerging Yellowtail Kingfish (*Seriola lalandi*) Aquaculture on the Chilean Coast, with Descriptions of Two New Species of *Paradeontacylix* (Aporocotylidae)

**DOI:** 10.3390/pathogens10070849

**Published:** 2021-07-06

**Authors:** Fabiola A. Sepúlveda, Luis A. Ñacari, Maria Teresa González

**Affiliations:** 1Instituto de Ciencias Naturales Alexander von Humboldt, Facultad de Ciencias del Mar y Recursos Biológicos, Universidad de Antofagasta, Angamos 601, Antofagasta 1240000, Chile; fabiolasepu@gmail.com; 2Programa Doctorado en Ciencias Aplicadas, Facultad de Ciencias del Mar y Recursos Biológicos, Universidad de Antofagasta, Angamos 601, Antofagasta 1240000, Chile; LUIS.NACARI.ENCISO@ua.cl

**Keywords:** aporocotylids, host specificity, molecular analyses, morphology, new species

## Abstract

Blood flukes are digeneans that infect wild and farmed fish that can cause a severe and potentially lethal disease in farmed fish. These parasites are undetectable in the larval stage based on macroscopic observations in the definitive host with the infection becoming evident when eggs accumulate in the branchial vessels. There are nine known species of the genus *Paradeontacylix* and seven exclusively parasitize *Seriola* spp. from several geographical areas. *Seriola lalandi* aquaculture farms are emerging at various localities in northern Chile. Here, we report, for the first time, two blood fluke species parasitizing *S. lalandi* in the Southeastern Pacific (Chile). In the laboratory, the gills and heart of fish were removed. The retained blood flukes were separated according to the infection site, fixed in 70% or 95% ethanol for taxonomic and molecular analysis, respectively. Morphometrical differences among the fluke species were evaluated with a principal component analysis (PCA) using proportional body measurements. Phylogenetic trees were constructed based on 28S rDNA, *cox*1 mDNA using Bayesian inference (BI), and maximum likelihood (ML). Based on morphology, morphometry, and molecular analyses, two new species are proposed: *P. humboldti* n. sp. from the gills and *P. olivai* n. sp. from the heart of *S. lalandi.* Both were clearly distinguished from other species of *Paradeontacylix* by a combination of morphologic features (posterior tegumental spines, testes arrangement, body size). The genetic distance (based on *cox*1) among species was >10%. *P. humboldti* n. sp. and *P. olivai* n. sp. are sister species (with a common ancestor) independent of *P. godfreyi* from *S. lalandi* in Australia. The newly identified parasites may pose a risk to farmed *S. lalandi* as aporocotylids have been the cause of diseases in farmed fish from other geographical areas. In addition, some cages of *S. lalandi* are currently maintained in an open circulating system, which could favor the transmission of these parasites (if involved hosts are present in the environment).

## 1. Introduction

Blood flukes are digeneans that infect the circulatory systems of wild and farmed fish [1,2,3] and can cause a severe disease in farmed *Seriola* spp. [4,5,6,7]. The disease occurs because the eggs accumulate in the gill filaments, leading to gill hyperplasia, egg encapsulation in the gills and ventricle, and papillae formation due to endothelial proliferation in the afferent branchial arteries [8]. In addition, hatching miracidia may cause multiple lesions and microhemorrhages, which in turn can trigger an inflammatory response and result in anemia [5]. This is especially the case when a massive hatch occurs, as has been reported for *Sanguinicola inermis* in *Cyprinus carpio* [9] and *Cardicola* sp. in *Sparus aurata* [10]. This pathology could be harmful in the aquaculture industry due this infection becoming evident when eggs accumulate in the branchial vessels and the fish become moribund [11]. Dead fish are characterized by an opened mouth and opercula, showing typical sign of suffocation [12]. The disease has been reported to be responsible for important losses regarding *Seriola dumerili* aquaculture, reaching 50–80% of the mortalities among fish in the 0+ and 1+ classes [5,12,13].

Blood flukes are restricted to a limited range of closely related definitive hosts. To date, nine aporocotylid species of the genus *Paradeontacylix* have been reported to almost exclusively parasitize *Seriola* spp. (Carangidae) in at least four geographical areas: the northwest Atlantic, northwest Pacific, southwest Pacific, and Mediterranean [4,14,15,16,17]. Seven of the nine species of *Paradeontacylix* infest *Seriola* spp.: *P. sanguinicoloides* parasitizes wild *Seriola dorsalis* (as *Seriola lalandi)* in Florida (northwest Atlantic [14]) but corresponding *S. dorsalis* [18], *P. grandispinus*, and *P. kampachi* parasitize farmed *S. dumerili* in Japan (northwest Pacific [4]); *P. godfreyi* parasitizes wild *S. lalandi* in Australia (Indian Pacific Ocean [17]); *P. balearicus* and *P. ibericus* parasitize wild *S. dumerili* on the Balearic Islands and Iberian Peninsula, respectively (Mediterranean); and *P. buri* parasitizes farmed *S. quinqueradiata* in Japan (northwest Pacific [16]). By contrast, *P. megalaspium* and *P. odhneri* have been reported to parasitize fish species other than the genus *Seriola*, e.g., *Megalaspis cordyla* (Carangidae) and *Takifugu porphyreus* (Tetraodontidae), respectively [19,20]. However, due to its atypical morphologic traits, it has been suggested that *P. odhneri* does not belong to *Paradeontacylix* [15,16].

For the emerging aquaculture industry to become sustainable and successful, it is essential to identify risk factors [21]. For this, the detection and correct identification of pathogenic parasites is an important first step [3,22]. *Seriola* spp. are successfully cultivated around the world, but not without diseases [21,23,24]. Although mass mortalities caused by *Paradeontacylix* spp. have been reported among farmed *S. dumerili* [4,12] and *S. quinqueradiata* [23] in Japan, and among farmed *S. dumerili* [5,13] in the Mediterranean, there are no records of this disease among farmed *S. lalandi* [17]. On the southeast Pacific coast, *S. lalandi* is one of the most important fish candidates for aquaculture diversification in Chile, and aquaculture activities have already begun at various localities in northern Chile [25,26,27]. The experimental cultures of yellowtail kingfish began in facilities of the Universidad de Antofagasta in the year 2000. Three other experimental cultures started in the last ten years (Univ. Tarapaca, Univ. Arturo Prat, and Chile Foundation UCN). The only company that cultivates fish commercially began its productive operations in 2008. This company maintains a mixed culture system of wild and farmed native specimens with tanks conditioned with a recirculating aquatic system. All the cultures in the experimental phase capture fish from the environment to maintain and to increase the number of brood fish.

Moderate-to-heavy infections involving the parasites *Zeuxapta seriolae* [28], *Benedenia humboldti* [27,29,30], and *Caligus lalandei* [27] have been reported in wild and farmed *Seriola lalandi* from the Chilean coast. Additionally, *Neobenedenia* sp. has only been reported in farmed *S. lalandi* [31] and there are not previous reports of blood flukes in wild or captive *S. lalandi*. However, recently, one dead *S. lalandi* from the experimental facility of Univ. Antofagasta was necropsied, and blood flukes were recovered from the heart and branchial arteries and numerous eggs were found in the branchial filaments. The parasites were identified as species of *Paradeontacylix*, indicating that the fish death was due to the parasites (pers. obs.). Consequently, the objective of this study was to report, for the first time, blood flukes infecting wild and farmed *Seriola lalandi* (experimental culture) from the Chilean coast (southeastern Pacific). Additionally, based on the differences in morphologic, morphometric, and molecular characteristics of the blood flukes obtained in this study compared to known species, two new species of *Paradeontacylix* are proposed and described here (*Paradeontacylix humboldti* n. sp. and *Paradeontacylix olivai* n. sp.).

## 2. Results

All specimens collected in this study were morphologically identified as *Paradeontacylix* spp. From here on, those will be referred to as *Paradeontacylix olivai* n. sp. and *Paradeontacylix humboldti* n. sp. The specimens *P. olivai* n. sp. were obtained from the heart and they were bigger than specimens *P. humboldti* n. sp. obtained from the blood vessels (Figure 1). *P. olivai* n. sp. was present in wild and farmed *S. lalandi* with an intensity of infection varying between 1 and 5 worms per fish heart (mean intensity = 2.3; prevalence = 20%). All of them were adults with eggs. About 39 eggs of *P. olivai* were recovered from gill filaments of farmed fish (Appendix A). *P. humboldti* was present only in farmed fish with an intensity of infection of 20 worms recovered from an afferent branchial artery.

### 2.1. Morphometric Analysis

PCA analysis showed that PC1, involving body width, male genital pore–posterior end distance, ovary length, ovary width, oviducal seminal receptacle length, oviducal seminal receptacle width, oötype length, and oötype width, explained 40% of the total variance. PC2, involving number of testes/BL and female genital pore–posterior end distance/BL, explained 18% of the total variance. Together, PC1 and PC2 explained 58% of the variance. *P. kampachi* and *P. ibericus* were in one group, and there was some overlap between *P. sanguinicoloides* and *P. godfreyi*. However, *P. grandispinus*, *P. balearicus*, and the two new species identified in this study were clearly different from each other (Figure 2).

### 2.2. Molecular Analysis

The unique sequences obtained in this study were coded with the following access numbers: MW599287-MW599288 (28S LSU rDNA) and MW598468-MW598470 (*cox*1 mDNA) (Table 1). Three sequences of *P. humboldti* n. sp. and five sequences of *P. olivai* n. sp. were obtained for each gene: 28S LSU rDNA (902 bp in length) and the *cox*1 mDNA (743 bp in length), respectively (Table 1). Regarding the 28S LSU rDNA, there was no intraspecific polymorphic sites for either species but there were 11 polymorphic sites between the new candidate species (*P. humboldti* n. sp. and *P. olivai* n. sp.). Regarding the *cox*1 gene, there were no intraspecific polymorphic sites among *cox*1 sequences of *P. humboldti* n. sp. while only one intraspecific polymorphic site was detected for *P. olivai* n. sp. and 67 polymorphic sites were detected between the new candidate species.

The final alignment of all datasets (including the *Paradeontacylix* spp. sequences from GenBank) resulted in 910 bp for 28S LSU rDNA and 416 bp for *cox*1 mDNA. Phylogenetic reconstructions based on the total molecular evidence (910 bp + 416 bp = 1326 bp) resulted in the same general topology for both inference methods (ML and BI) (Figure 3). The species of *Paradeontacylix* were clustered into a single monophyletic clade, which was strongly supported by a high posterior probability in the BI analysis (pp = 1) and by the bootstrap support value in the ML analysis (bML = 99) (Figure 3). In each phylogenetic tree there were three subclades with moderate-to-strong statistical support: one involved *P. humboldti* n. sp. and *P. olivai* n. sp., the second clade (which is a sister clade of the new candidate species) involved *P. balearicus* and *P. grandispinus*, and the third clade involved *P. ibericus* and *P. kampachi* (Figure 3). *P. godfreyi* appeared as a basal species within the genus *Paradeontacylix*.

Based on *cox*1 mDNA, *P. humboldti* n. sp. and *P. olivai* n. sp. had genetic distances >10% between them and between them and the other species of *Paradeontacylix.* The pairwise sequence divergences for each of the two molecular markers are shown in Table 2.

### 2.3. Morphologic Descriptions

#### 2.3.1. *Paradeontacylix humboldti* n. sp.

##### Description

Host: *Seriola lalandi* Valenciennes, 1833.

Locality: San Jorge Bay (23°35′ S, 70°25′ W), off Antofagasta, northern Chile.

Location in host: Afferent branchial artery.

Representative sequences: GenBank accession numbers, partial 28S LSU rDNA MW599287; *cox*1 mDNA MW598468.

Etymology: The species is named in honor of German naturalist Alexander von Humboldt.

Specimens: Holotype and one paratype have been deposited in the MNHNCL, encoded as: PLAT-15021 and PLAT-15022, respectively.

Based on two whole-mounted gravid adult specimens. Body measurements are given in Table 3. Body smooth, elongated, dorsoventrally flattened, lancet-shaped (Figure 4A). Longer than wide by 12–13 (12.5, n = 2) times. Marginal tegumental spines ventrolateral, 286–436 (361, n = 4) rows on either side of body; same size throughout body, averaging 4 long by 1 wide (maximum width at spine base), 4–9 per row, decreasing in number toward both extremities to about 4 at anterior and posterior ends. Posteriorly, 9 large tegumental spines, conspicuous, claw-like distally, arranged in 4 longitudinal rows each comprising 2–3 spines (Figure 4C). Additionally, medium-sized posterior tegumental spines, arranged on either side of large spines in 3 rows (Figure 4C).

Nerve commissure 110–131 (121, n = 2) from anterior end. Mouth opening ventro-subterminally, 28 (n = 2) from anterior end. Esophagus sinuous, gradually widening toward its end, surrounded by conspicuous gland cells positioned at 342–451 (396, n = 2) from oesophagus anterior end. Short anterior and elongate posterior intestinal caeca forming H-shaped; anterior caeca 47–48 (48, n = 4) long; posterior caeca 688–878 (783, n = 4) long, terminating blindly among testicular field, anterior to ovary (Figure 4A).

Testes arranged irregularly between posterior caeca, round or ellipsoidal. Testicular zone occupying 33% (n = 2) of body length (Figure 4A). Vas deferens starting from posterior testes, passing ovary dorsally, following curved path to form seminal vesicle filling entire cirrus sac (Figure 4B). Seminal vesicle enclosed in cirrus sac (Figure 4B). Cirrus pouch thin-walled (Figure 4B). Male genital pore dorsal, 12–24 (18, n = 2) from sinistral body margin, 257–349 (303, n = 2) from posterior end of body (Figure 4B).

Ovary oval-shaped, posterior to testes, ventral to vas deferens, corresponding to 25–26% (25.5%, n = 2) of body length (Figure 4B). Oviduct originating at posterior end part of ovary. Expanding in the middle of oviduct is form oviducal seminal receptacle (Figure 4B). Oviducal seminal receptacle narrowing progressively and joining common vitelline duct before forming oötype (Figure 4B). Oötype slightly on its dextral side, 202–236 (219, n = 2) from posterior end of body, surrounded by Mehlis’ gland (Figure 4B). Vitelline duct passing posteriorly, sinistral to oviduct. Uterus occupying zone from level of posterior end of ovary to level slightly behind oötype, sinistral to common vitelline duct, after ascending through several coils (filling space immediately posterior to ovary) and then descending to female genital pore (Figure 4B). Female genital pore dorsal, opening on sinistral side, antero-sinistral to male genital pore, 28–31 (30, n = 2) from sinistral body margin, 145–208 (177, n = 2) from male genital pore, 409–548 (478, n = 2) from posterior end of body. Eggs ellipsoidal (Figure 4B). Excretory vesicle and pore not observed. Vitellarium extending from level of nerve commissure to level of posterior end of ovary (Figure 4A).

##### Remarks

The *P. humboldti* n. sp. displays all the diagnostic characteristics of the genus *Paradeontacylix* McIntosh, 1934. Here, we have initially classified it as a species of the genus *Paradeontacylix* according to testes distribution as irregular (random, without a pattern) and regular (in two well-defined rows). Based on this criterion, the first group includes *P. grandispinus*, *P. balearicus*, *P. sanguinicoloides*, and *P. godfreyi* with irregular testes distribution, while a second group includes: *P. kampachi*, *P. buri*, *P. ibericus*, and *P. megalaspium*, which present regular distribution. The distribution of testes in the new species *P. humboldti* n. sp. is irregular. Additionally, it differs significantly in the number of testes, intestine shape, and ovary shape from *P. grandispinus* (63–69 vs. 19–32 testes; H-shaped vs. X-shaped intestine; oval-shaped vs. heart-shaped ovary, respectively) and *P. balearicus* (63–69 vs. 20–26 testes; H-shaped vs. X-shaped intestine; oval-shaped vs. shield-shaped ovary, respectively). It also differs from *P. sanguinicoloides* in the shape of the large posterior tegumental spines (claw vs. rose thorn, respectively), ovary shape (oval-shaped vs. heart-shaped, respectively) and position of female genital pore (antero-sinistral vs. antero-mesal of male genital pore, respectively). *P. humboldti* n. sp. differs from *P. godfreyi* in the ovary shape (oval-shaped vs. heart-shaped) and testes position in the body (middle and posterior vs. posterior third). It is also approximately half of the maximum body widths of *P. sanguinicoloides* (155–179 vs. 330, respectively) and *P. godfreyi* (155–179 vs. 357–566, respectively). *P. humboldti* n. sp. is half the body length of *P. godfreyi* (1858–2352 vs. 3739–4215, respectively). Additionally, the position of the female genital pore, which opens without crossing vas deferens, is only shared with *S. godfreyi*.

#### 2.3.2. *Paradeontacylix olivai* n. sp.

##### Description

Host: *Seriola lalandi* Valenciennes, 1833.

Locality: San Jorge Bay (23°35′ S, 70°25′ W), off Antofagasta, northern Chile.

Location in host: Heart.

Prevalence: 20%.

Representative sequences: GenBank accession numbers: partial 28S LSU rDNA MW599288; *cox*1 mDNA MW598469-70.

Etymology: The species is named in honor of Dr. Marcelo Oliva, who has studied marine parasites for over 40 years in Chile.

Specimens: Holotype and one paratype have been deposited in the MNHNCL, encoded as: PLAT-15019 and PLAT-15020, respectively.

Based on eight whole-mounted adult specimens. Body measurements are given in Table 3. Body smooth, elongated, dorsoventrally flattened, lancet-shaped (Figure 5A). Longer than wide by 11–18 (13, n = 8) times. Marginal tegumental spines ventrolateral, 478–650 (563, n = 6) rows on either side of body; same size throughout body, 3–14 per row, decreasing in number toward both extremities to about 3 at anterior end and 3–4 at posterior end. Posteriorly, 13–17 large tegumental spines, conspicuous, claw-like distally, arranged in 4 rows each comprising 3–5 spines (Figure 5C). Additionally, medium-sized posterior tegumental spines arranged on either side of large spines in 3 or 4 rows (Figure 5C).

Nerve commissure 137–209 (166, n = 7) from anterior end. Mouth opening ventro-subterminally, 8–22 (16, n = 7) from anterior end. Esophagus sinuous, 14–30% (25%, n = 7) of body length, gradually widening toward its end, surrounded by conspicuous gland cells positioned at 697–940 (802, n = 6) from oesophagus anterior end. Short anterior and elongate posterior intestinal caeca forming X-shaped; anterior caeca and posterior caeca terminating blindly among testicular field, anterior to ovary (Figure 5A).

Testes arranged regularly in two rows between posterior extremities of caeca, round or ellipsoidal. Testicular field occupying 33–40% (36%, n = 8) of body length (Figure 5A). Vas deferens starting from posterior testes, passing dorsally to ovary, following a curved path to form seminal vesicle (Figure 5B). Seminal vesicle enclosed in cirrus sac (Figure 5B). Cirrus pouch cylindrical, thin-walled (Figure 5B). Male genital pore dorsal, 20–34 (27, n = 7) from sinistral body margin, 429–542 (496, n = 7) from posterior end of body (Figure 5B).

Ovary shield-shaped, distally elongated, posterior to testes, ventral to vas deferens, 778–1100 (911, n = 8) from its posterior end to posterior end of body, corresponding to 21–23% (22%, n = 8) of body length (Figure 5B). Oviduct originating at dextral posterior end part of ovary. Posterior part of oviduct expanding to form oviducal seminal receptacle (Figure 5B). Oviducal seminal receptacle narrowing progressively and joining common vitelline duct before forming oötype (Figure 5B). Oötype in mid-line, 330–443 (361, n = 6) from posterior end of body, surrounded by Mehlis’ gland (Figure 5B). Vitelline duct passing posteriorly, sinistral to oviduct. Uterus occupying zone from posterior end of ovary to oötype, sinistral to common vitelline duct, after ascending through several coils (filling space immediately posterior to ovary) and then descending to female genital pore (Figure 5B). Female genital pore dorsal, opening in midline, antero-dextral to male genital pore, 102–174 (143, n = 5) from sinistral body margin, 167–204 (190, n = 5) from male genital pore, 616–753 (664, n = 5) from posterior end of body. Eggs intrauterine, ellipsoidal (Figure 5B). Excretory vesicle and pore not observed. Vitellarium extending from level at or near nerve commissure to level of posterior end of ovary (Figure 5A).

##### Remarks

The distribution of testes in *P. olivai* n. sp. is regular, as they are distributed in two rows as in *P. kampachi*, *P. buri*, *P. ibericus*, and *P. megalaspium*. However, *P. olivai* n. sp. differs from *P. buri* as the latter does not have large posterior tegumental spines. *P. olivai* n. sp. also has fewer testes as well as fewer and smaller large posterior tegumental spines than *P. kampachi* (41–45 vs. 50–71 testes; ([3–5 spines] × 4 rows) vs. ([2–7 spines] × [5–12 rows]) with length of 15–39 vs. 11–15, respectively). It also differs from *P. ibericus*, principally in the number and size of large posterior tegumental spines (([3–5 spines] × 4 rows) vs. ([2–3 spines] × 5 rows) with mean length of 24 vs. 13, respectively) and ovary shape (shield-shaped vs. kidney-shaped). Lastly, it differs from *P. megalaspium* by a combination of characters based on presence of large posterior tegumental spines (present vs. absent, respectively), maximum number of rows of transversal marginal tegumental spines (14 vs. 10, respectively), number of testes (41–45 vs. 70–86, respectively) and intestine shape (X-shaped vs. H-shaped, respectively).

*P. olivai* n. sp. differs from *P. humboldti* n. sp. regarding the following traits: number of medium-sized posterior tegumental spines (([2–4 spines] × 3 rows) vs. (3 spines × 3 rows), respectively), distribution of testes (regular, in two rows vs. irregular, respectively), number of testes (43 vs. 66, respectively), ovary shape (shield-shaped vs. oval-shaped, respectively), intestine shape (X-shaped vs. H-shaped, respectively) and body length (3567–5214 vs. 1858–2353, respectively).

## 3. Discussion

The detection and correct identification of potential pathogenic parasites are an important as a first step to support emergent aquaculture [3,22]. The combination of morphologic and molecular characteristics is a strong and reliable method to identify species [32]. The conservative 28S LSU gene is an efficient marker for analyzing the phylogeny of digeneans at taxonomic levels, such as the genus and family [33,34], while the use of a mitochondrial marker as a DNA barcode has been useful for species discrimination [15,35]. Furthermore, using two or more independent loci (as in this study) provides advantages when estimating species-level relationships and testing hypotheses regarding species delimitation [36]. By analyzing new DNA sequences from *P. humboldti* n. sp. and *P. olivai* n. sp., our analysis complemented the previous analyses on the phylogenetic relationships between species of *Paradeontacylix* reported [15,16]. We found that *P. godfreyi* from *S. lalandi* in the Indian Pacific Ocean, south Australia, is located at the basal position within the *Paradeontacylix* spp. clade with strong nodal support. However, this result could be related to the genetic marker because previous authors used ITS-2 fragment, which is more variable than 28S gene (domains C1–D2). Regardless of this difference in tree topology, previous authors [15,16] demonstrated absence of influence of the host-phylogeography on the phylogenetic relationship of *Paradeontacylix* spp. in *S. dumerili*. More precisely, *P. grandispinus* (found in Japan) and *P. balearicus* (found in the Mediterranean) are genetically related as well as *P. kampachi* (found in Japan) and *P. ibericus* (found in the Mediterranean). Additionally, we found that the two new species, *P. humboldti* n. sp. and *P. olivai* n. sp. from *S. lalandi* in the southeastern Pacific, belong to the same (monophyletic) clade and represent a sister clade of *P. balearicus* and *P. grandispinus*. This suggests that they underwent a relatively recent divergence within the phylogeny of the genus *Paradeontacylix*. These results, however, must be confirmed using other more resolutive molecular markers.

The final host fish of *Paradeontacylix* spp., such as *S. dumerili* and *S. lalandi*, are known to be long-distance migratory fish species with genetically structured populations across their extensive geographical distributions [37,38,39,40]. Two populations of *S. dumerili* occur in the northeast Atlantic [39] and two other populations occur in the northwest Pacific [37]. This suggests that populations of this host species have undergone genetic divergence in the past as a consequence of historical processes [41,42] and that the ancestral parasite species had existed before the separation of the *S. dumerili* populations between Japan and the Mediterranean [15]. This would explain how a host fish species can come to harbor a pair of genetically related *Paradeontacylix* species (e.g., *S. dumerili* harbors *P. grandispinus* (in Japan) and *P. balearicus* (in the Mediterranean), and *S. dumerili* also harbors *P. grandispinus* (in Japan) and *P. balearicus* (in the Mediterranean)), which are reported to be restricted by the current geographical distribution of their host species populations. Similarly, there are at least four genetically distinct *S. lalandi* populations worldwide [18,40] with a single population of *S. lalandi* distributed in the south Pacific [40]. However, the spatio-temporal genetic structure for *S. lalandi* from the southeastern Pacific coast (the same area as in the present study) [43] and two populations of *S. lalandi* (on the Australian and New Zealand coasts) have been reported [44]. This could explain the existence of the two new species of *Paradeontacylix* in *S. lalandi* in the southeastern Pacific (*P. humboldti* n. sp. and *P. olivai* n. sp.), which differ from the species *P. godfreyi* recorded in *S. lalandi* in the Indian Pacific [17]. Therefore, further studies are required to clarify whether one or more *Paradeontacylix* species parasitize *S. lalandi* on the Australian coast and to know whether this parasite presents high or low host specificity as suggested by Hutson and Whittinton [17].

All marine aporocotylids whose life cycles are known use terebellid polychaetes (*Nicolea gracilibranchis*, *Longicarpus modestus*, and *Reterebella aloba*; *Terebella* sp.; *Neoamphitrite vigintipes)* as intermediate hosts [6,45,46]. However, the intermediate hosts for *Paradeontacylix* spp. are unknown, but it has been suggested that, for aporocotylids, direct penetration by cercariae is the dominant infection route in fishes, i.e., the infection could be independent of the host diet [3]. This is critical information as it implies that the life cycle and infection potential of the parasite may be independent of the trophic web. Currently, in northern Chile the emerging aquaculture capture wild fish to improve the genetic variability of the brood fish, and already at least one dead fish has evidenced infection by *Paradeontacylix* spp. This means that if the parasite exists in the environment (in intermediate hosts and wild fishes), and if there is an open aquaculture system in the region, there is a high probability that the captive fish will acquire the parasite. In the practice, blood flukes of fish are difficult to detect as they infest the host vascular system [3,47]. Montero et al. [48] reported, for the first time, that although *P. ibericus* infections in farmed *S. dumerili* were undetectable based on macroscopic observation, there were encysted cercariae (schistosomula) in fixed histological sections of muscle, which could be useful for early diagnosis of this pathology. After experimentally evaluating the parasite development, they reported that small juvenile *P. ibericus* worms were still found in the muscles and lymphatic system >100 days after the transfer of the fish to tanks, and *P. ibericus* adults (with a recognizable reproductive system) were recovered 8 months after the transfer. There was a higher intensity of *P. ibericus* in the girdle muscles, head kidney, and sinus venosus (involving both juveniles and adult worms) while a low intensity (involving only adults) was detected in the gills. In our study, wild fish were infected mainly with *P. olivai* adults detected in the heart while farmed fish were infected with both species, although predominantly with *P. humboldti*, and numerous eggs were observed in the gills. Further studies focused to evaluate infection levels, proportion of each species, presence/absence of eggs in the gill, or heart and their variability in the sampling periods will be necessary to elucidate the early biological traits and life cycles of the two new blood fluke species infesting *S. lalandi* in the southeastern Pacific as it was done by Montero et al. [48].

As different parasites species can have different infection patterns and in turn respond differentially to control treatments, the identification of the two new species of *Paradeontacylix* described here is crucial to develop a diagnosis protocol of these potential pathogens in farmed fish, which should be accompanied by records on the occurrence of these parasites, monitoring of the fish condition, and mortality due to any of these parasites. This protocol will allow the implementation of preventive management and control of this potential disease in the emerging fish farming industry in Chile.

## 4. Material and Methods

### 4.1. Sample Collection

Twenty specimens of *S. lalandi* ranged between 46 and 84 cm fork length were acquired at a fish market in Antofagasta (24° S), Chile, captured by an artisanal fishery in the nearshore area (24° S–26° S) during the summer season (January–February) in 2018 and 2019. In these months, wild *S. lalandi* migrate to the Chilean coast when the water temperature (17 °C–21 °C) increases [43]. Additionally, one specimen (about 4 kg) of *S. lalandi*, captured in the summer of 2017 and maintained captive in the hatchery (with an open circulatory marine water intake and effluent system) of the Univ. Antofagasta was found dead and infected with blood flukes in August 2019. The fish was characterized as having an opened mouth and opercula, showing typical signs of suffocation. In the laboratory, the gills and heart of the fish were removed. Each heart was opened and washed in a Petri dish with freshwater, which was then filtered with a sieve. Each gill arch was dissected longitudinally and washed following the same protocol. The retained blood flukes were carefully separated according to the infection site (heart or gill) and fixed in 70% or 95% ethanol for taxonomic identification and molecular analysis, respectively. Parasitological indexes (mean intensity and prevalence) were calculated [49].

### 4.2. Morphological Description and Morphometrical Analyses

The flukes were stained with acetocarmine or Gomori’s trichrome, dehydrated in ethanol (70–100%), cleared with clove oil (Sigma-Aldrich, Taufkirchen, Germany), and mounted in Eukitt^®^ mounting medium (O. Kindler GmbH, Freiburg, Germany). The flukes were photographed (M125 camera; Leica, Wetzlar, Germany) and measured using the ImageJ software [50]. Measurements were made in micrometers (µm) and are given as the range, followed by the mean and the number of structures measured or counted in parentheses. Specimens were drawn using a compound microscope with a drawing tube. The type material was submitted to Museo Natural de Historia Natural (MNHNCL) in Santiago, Chile.

Morphometric analyses involved comparing the measurements of the specimens in this study with the measurements of *P. godfreyi*, *P. sanguinicoloides*, *P. grandispinus*, *P. kampachi*, *P. balearicus*, and *P. ibericus* obtained from original publications (Table 2). For this, we used the measurements (minimum and maximum) directly reported in the publications while other measurements were estimated from the drawings. *P. buri* and *P. megalaspium* were not considered because these species do not have large posterior tegumental spines. The source and number of specimens per host species and geographical area are given in Table 2.

To evaluate differences in morphometry among the fluke species, a principal component analysis (PCA) was performed using proportional body measurements as all relevant fluke measurements are correlated with body length [51]. For this analysis, the following 16 body measurements divided by total body length (BL) were used: maximum body width, number of marginal tegumental spine rows, posterior spine length, esophagus length, anterior caeca–intestine distance, posterior caeca–intestine distance, number of testes, testicular area length, male genital pore–posterior end distance, ovary length, ovary width, oviducal seminal receptacle length, oviducal seminal receptacle width, oötype length, oötype width, and female genital pore–posterior end distance. These analyses were performed using the Statistica 7.0 software (StatSoft Inc., Tulsa, OK, USA).

### 4.3. Molecular Analysis

#### 4.3.1. DNA Extraction and Amplification

The DNA was isolated following a modified version of a protocol reported in [52]. This involved treatment with sodium dodecyl sulphate, digestion with proteinase K, NaCl protein precipitation, and subsequent ethanol precipitation. The DNA was eluted in nuclease-free water and quantified using a BioSpec-nano spectrophotometer (Shimadzu, Japan).

For the molecular analyses, regions within the 28S ribosomal DNA large subunit (LSU rDNA) and the mitochondrial cytochrome *c* oxidase 1 gene (*cox*1 mDNA) were amplified by polymerase chain reaction (PCR). LSU rDNA was amplified using the forward primer C1 (5′-ACCCGCTGAATTTAAGCAT-3′) and the reverse primer D2 (5′-TGGTCCGTGTTTCAAGAC-3′) [53]; *cox*1 mtDNA was amplified using the forward primer JB3 (5′-TTTTTTGGGCATCCTGAGGTTTAT-3′) and the reverse primer *COX*1 (5′-AATCATGATGCAAAAGGTA-3′) [54]. Each PCR reaction was carried out in a final volume of 35 μL comprising five standard units of GoTaq DNA polymerase (Promega), 7 μL of 5 × PCR buffer, 5.6 μL of MgCl_2_ (25 mM), 2.1 μL of BSA (10 mg/mL), 0.7 μL of deoxynucleotide triphosphate (dNTP; 10 mM), 10 pM of each primer, 3 μL of template DNA, and sufficient nuclease-free H_2_O to make the total volume up to 35 μL. A Boeco Ecogermany M-240R Thermal Cycler (Boeckel, Hamburg, Germany) was used to carry out PCR for LSU rDNA and *cox*1 mDNA using the programs reported [53,54], respectively. The PCR products were sent to Macrogen (Seoul, Korea; http://www.macrogen.com, accessed on 15 December 2020) for purification and sequencing of both the DNA forward and reverse strands. The sequences were edited and contigs were assembled using ProSeq 2.9 beta [55]. All unique sequences obtained during this study were deposited into GenBank.

#### 4.3.2. Phylogenetic Reconstruction

The sequences obtained in this study were aligned with sequences of *Paradeontacylix* spp. obtained from GenBank using Clustal X [56] (Table 1). Visual inspection was then performed using ProSeq v2.91 [55] in order to edit the length of the final dataset. The jModelTest v0.1.1 tool [57] was used to identify the best evolutionary model for each gene. Gene concatenation (LSU + *cox*1) was performed using Mesquite v2.75 [58]. Phylogenetic trees were constructed based on 28S LSU rDNA, *cox*1 mDNA, and the concatenated genes using Bayesian inference (BI) and maximum likelihood (ML) analyses. Five members of the Aporocotylidae family, *Cardicola forsteri*, *C. opisthorchis*, *Psettarium nolani*, *P. sinense*, and *Plethorchis acanthus*, were selected as outgroups based on their close phylogenetic relationships with the genus *Paradeontacylix*. Sequences of the outgroup taxa were obtained from GenBank (Table 1).

The BI analyses were conducted using MrBayes [59] with the following parameters: nst = 6 and rates = invgamma according to the evolutionary model determined by jModeltest v0.1.1 for each gene (GTR + G + I for 28S LSU rDNA and TIM2 + G + I for *cox*1 mDNA, and replaced by GTR + G + I for MrBayes software). Each analysis was performed for 5,000,000 generations, with one run of four chains, sampling every 100 generations. Support for nodes in the BI tree topology was based on posterior probability. The initial 25% was discarded as burn-in. The results were visualized using TRACER v1.7 [60]. The ML analyses were performed using W-IQ-TREE (http://iqtree.cibiv.univie.ac.at/ [61] accessed on 15 March 2021), with 1000 bootstrap replicates for statistical support. The trees based on the concatenated genes were visualized and edited in FigTree v1.4.4 (http://tree.bio.ed.ac.uk/software/figtree/, accessed on 15 March 2021).

Finally, the pairwise p-distances for LSU rDNA and *cox*1 mDNA sequences among multiple species of *Paradeontacylix* were analyzed using the MEGA v6 software [62].

## 5. Conclusions

*P. humboldti* n. sp. and *P. olivai* n. sp. described in the present study constitute two new species of blood flukes that infest *Seriola lalandi*. The identity of each of the two new species is supported by morphological, morphometric, and molecular data.

## Figures and Tables

**Figure 1 pathogens-10-00849-f001:**
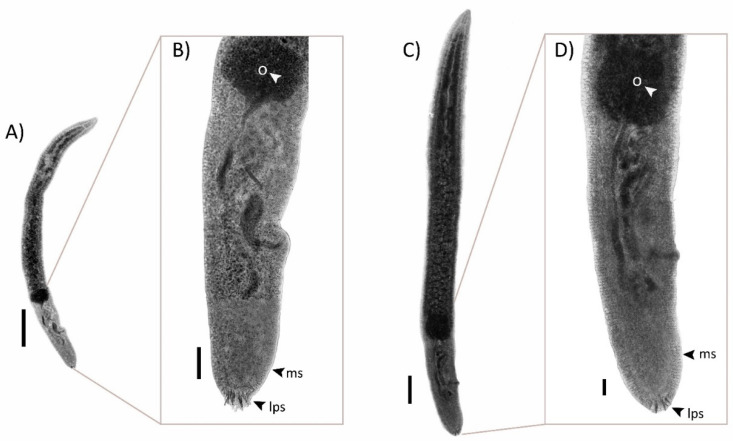
Light microscopy photographs of *Paradeontacylix* sp. from *Seriola lalandi* on the Chilean coast. (**A**) Ventral view of complete body of *Paradeontacylix humboldti* n. sp. and (**B**) details of posterior region. (**C**) Ventral view of complete body of *Paradeontacylix olivai* n. sp. and (**D**) details of posterior region. Abbreviations: lps, large posterior spines; ms, marginal spines; o, ovary. Scale bars: (**A**,**C**), 250 μm; (**B**,**D**), 50 μm.

**Figure 2 pathogens-10-00849-f002:**
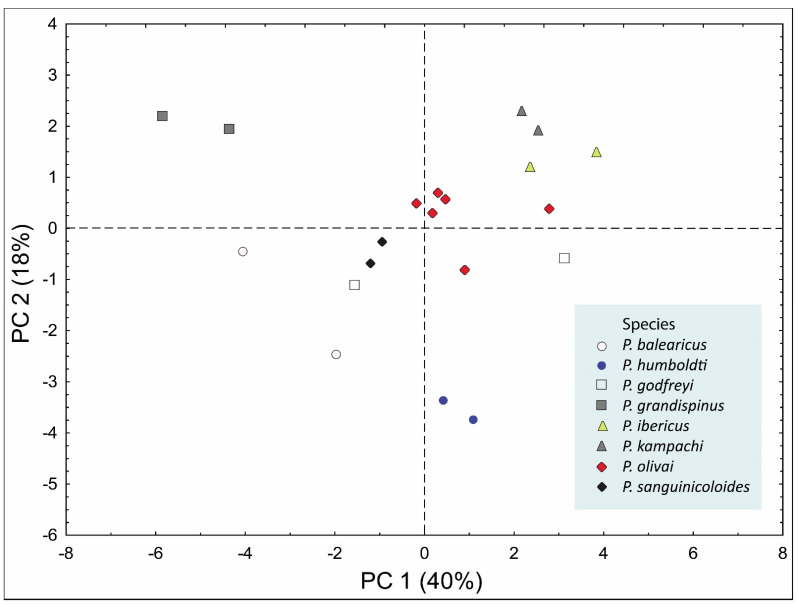
Results of multivariate analyses of proportional measurements of *Paradeontacylix* spp. Each symbol in the graph is defined in the box on the lower right.

**Figure 3 pathogens-10-00849-f003:**
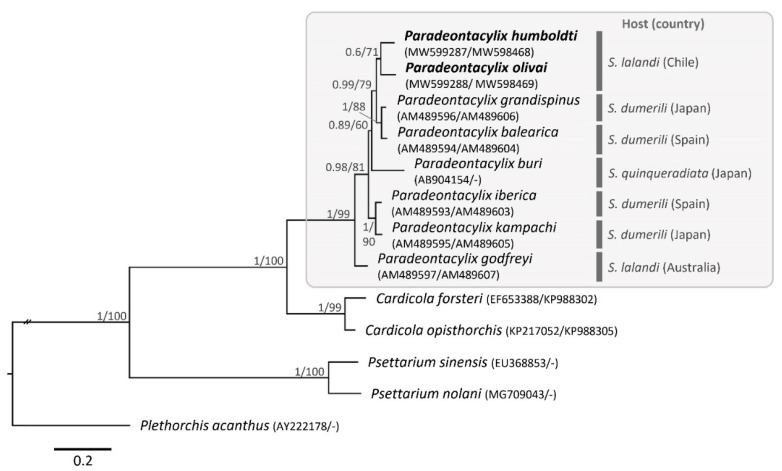
Phylogenetic tree based on partial 28S LSU rDNA and *cox*1 mDNA sequences of *Paradeontacylix* spp. Numbers at the nodes show posterior probability based on the Bayesian inference analysis/bootstrap support values based on 1000 replicates in the maximum likelihood analysis.

**Figure 4 pathogens-10-00849-f004:**
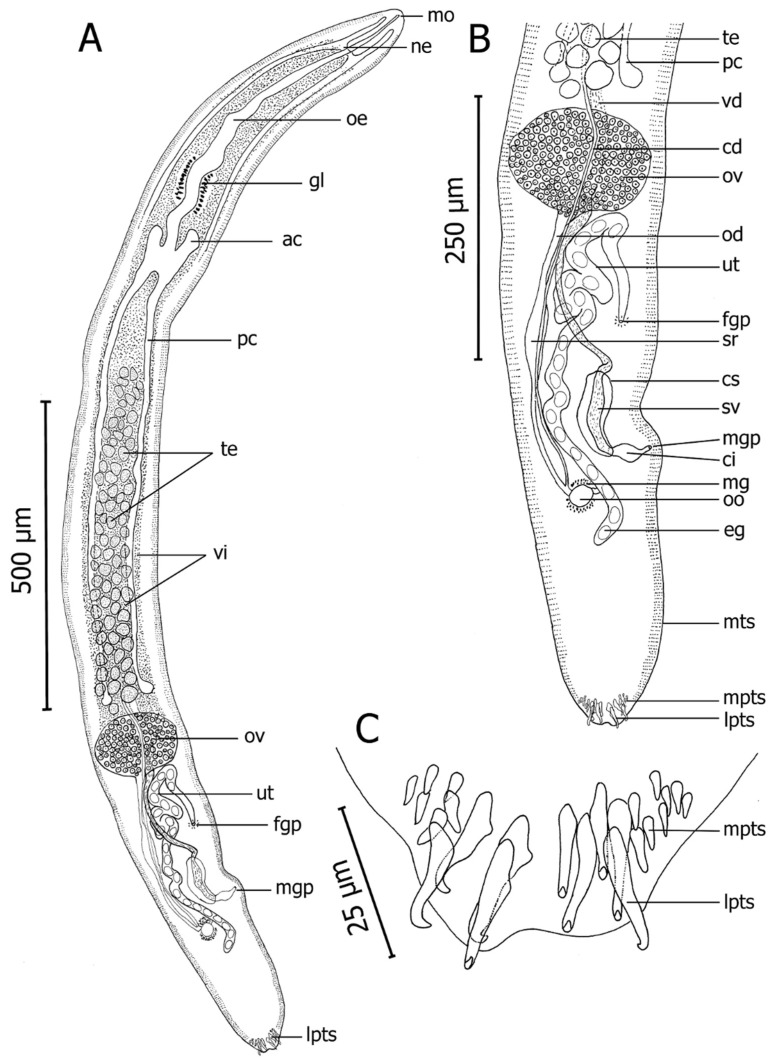
Holotype of *Paradeontacylix humboldti* n. sp. from afferent branchial artery of *Seriola lalandi*. (**A**) Ventral view of whole worm. (**B**) Ventral view of posterior part of body showing one part of testicular field and post-ovarian region. (**C**) Posterior tegumental spines showing large claw-like posterior tegumental spines. Abbreviations: ac, anterior caeca; cd, common vitelline duct; ci, cirrus; cs, cirrus sac; eg, eggs; fgp, female genital pore; gl, gland cells; lpts, large posterior tegumental spines; mg, Mehlis’ gland; mgp, male genital pore; mo, mouth; mpts, medium posterior tegumental spines; mts, marginal tegumental spines; ne, nerve commissure; od, oviduct; oe, oesophagus; oo, oötype; ov, ovary; pc, posterior caeca; sr, oviducal seminal receptacle; sv, seminal vesicle; te, testes; ut, uterus; vd, vas deferens; vi, vitellarium.

**Figure 5 pathogens-10-00849-f005:**
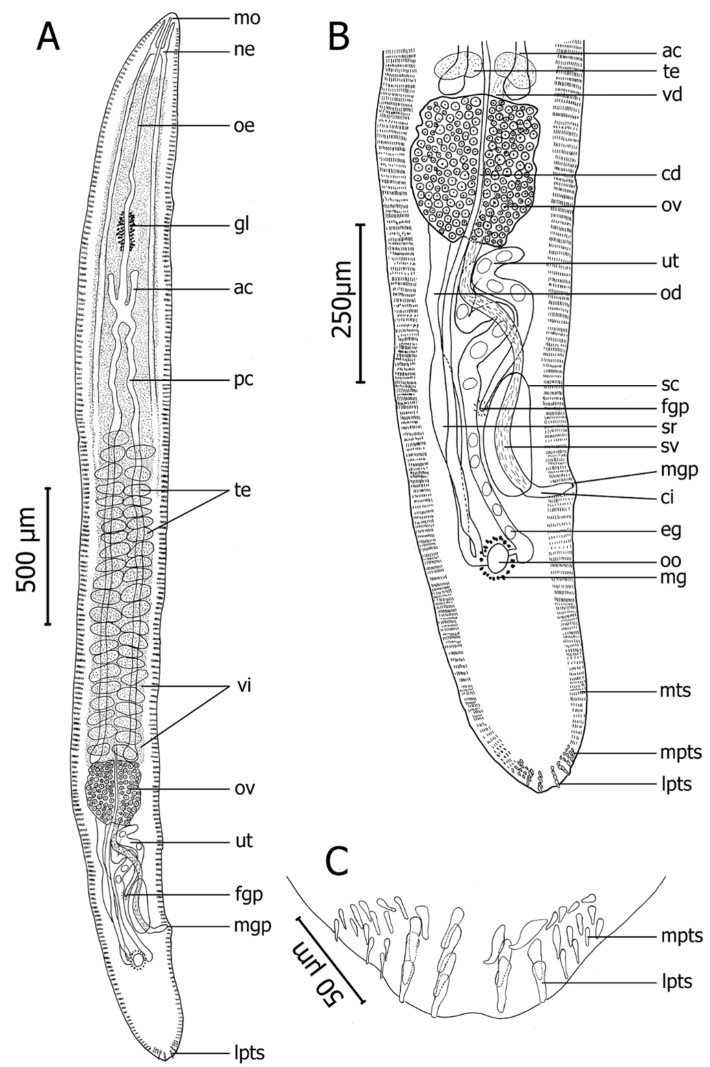
Holotype of *Paradeontacylix olivai* n. sp. from heart of *Seriola lalandi.* (**A**) Ventral view of whole worm. (**B**) Ventral view of posterior part of body, showing one part of testicular field and post-ovarian region. (**C**) Posterior tegumental spines, showing large claw-like posterior tegumental spines. Abbreviations as in Figure 4.

**Table 1 pathogens-10-00849-t001:** List of blood fluke species, host species, geographic reports, and GenBank accession numbers (nuclear and mitochondrial genes) used in this study. N, number of specimens belonging to different *Paradeontacylix* species used for morphometrical/molecular analyses.

Species	N	Host	Country	Access Number		Author(s)
				28S	*cox*1	
*P. balearicus*	2/1	*Seriola dumerili*	Spain	AM489594	AM489604	Repullés-Albelda et al., 2008
*P. buri*	-/1	*S. quinqueradiata*	Japan	AB904154	-	Ogawa et al., 2015
*P. godfreyi*	2/1	*S. lalandi*	Australia	AM489597	AM489607	Repullés-Albelda et al., 2008
*P. grandispinus*	2/1	*S. dumerili*	Japan	AM489596	AM489606	Repullés-Albelda et al., 2008
*P. humboldti*	2/3	*S. lalandi*	Chile	MW599287	MW598468	This study
*P. ibericus*	2/1	*S. dumerili*	Spain	AM489593	AM489603	Repullés-Albelda et al., 2008
*P. kampachi*	2/1	*S. dumerili*	Japan	AM489595	AM489605	Repullés-Albelda et al., 2008
*P. olivai*	8/5	*S. lalandi*	Chile	MW599288	MW598468-70	This study
*P. sanguinicoloides*	2/-	*S. dorsalis*	Florida	-	-	McIntosh, 1934
*Cardicola forsteri*	-	*Thunnus thynnus*	Spain	EF653388	KP988302	Aiken et al., 2007/Palacios-Abella et al., 2015
*Cardicola opisthorchis*	-	*Thunnus thynnus*	Spain	KP217052	KP988305	Unpublished/Palacios-Abella et al., 2015
*Psettarium nolani*	-	*Arothron hispidus*	Australia	MG709043	-	Yong et al., 2018
*Psettarium sinensis*	-	*Fugu rubripes*	China	EU368853	-	Unpublished
*Plethorchis acanthus*	-	*Mugil cephalus*	Australia	AY222178	-	Olson et al., 2003

**Table 2 pathogens-10-00849-t002:** Genetic distance between *Paradeontacylix* spp based on 28S LSU rDNA and *cox*1 mDNA, respectively. Lower half shows the percentage differences between the paired comparisons using 28S LSU rDNA (based on 910 bp) and upper half shows the percentage differences between the paired comparisons using *cox*1 (based on 416 bp).

	Species	1	2	3	4	5	6	7	8
1	*P. humboldti*		10.26	10.50	-	11.69	16.47	11.93	13.84
2	*P. olivai*	1.25		10.74	-	12.17	15.51	11.22	11.93
3	*P. grandispinus*	1.02	1.02		-	11.69	13.84	6.20	11.46
4	*P. buri*	2.39	2.39	2.05		-	-	-	-
5	*P. ibericus*	1.59	1.59	1.25	1.94		15.99	11.46	6.921
6	*P. godfreyi*	1.48	1.48	1.14	2.05	0.79		13.6	15.04
7	*P. balearicus*	1.02	1.02	0.23	2.05	1.25	1.14		11.46
8	*P. kampachi*	1.48	1.48	1.14	1.82	0.23	0.68	1.14	

**Table 3 pathogens-10-00849-t003:** Body measurements of *Paradeontacylix humboldti* n sp. and *Paradeontacylix olivai* n. sp infecting *Seriola lalandi* from Chile.

Body Measurement	*Paradeontacylix humboldti* n. sp.	*Paradeontacylix olivai* n. sp.
Body length	1858–2353 (2105, n = 2)	3567–5214 (4099, n = 8)
Body maximum width	155–179 (167, n = 2)	204–369 (324, n = 8)
Marginal tegumental spine rows	286–436 (361, n = 4)	478–650 (563, n = 6)
Marginal tegumental spine length	4	5–8 (6, n = 7)
Marginal tegumental spine width	1	1–2 (1.6, n = 7)
Marginal tegumental spine per row	[4–9]	[3–14]
No. of large posterior tegumental spines [No. spines × No. rows]	([(2–3) × 4]–[(2–3) × 4])	[(3–5) × 4]
Large posterior tegumental spines length	15–22 (19, n = 18)	15–39 (24, n = 30)
Large posterior tegumental spines width (maximum wide at spine base)	3–5 (4, n = 18)	4–7 (5, n = 30)
No. of medium posterior tegumental spines [No. spines × No. rows]	[3 × 3]	[(3–4) × 3]
Medium posterior tegumental spines length	7–13 (9, n = 12)	6–17 (11, n = 16)
Medium posterior tegumental spines width	1–2 (2, n = 12)	2–4 (3, n = 18)
Mouth opening from anterior end	28 (n = 2)	8–22 (16, n = 7)
Oesophagus length	533–660 (597, n = 2)	496–1268 (1040, n = 7)
Oesophagus (percentage of body length)	28–29% (n = 2)	14–30% (25%, n = 7)
Oesophagus’s glands cell from anterior end length	342–451 (396, n = 2)	706–962 (814, n = 6)
Intestine anterior caeca length	47–48 (48, n = 4)	60–126 (103, n = 14)
Intestine posterior caeca length	688–878 (783, n = 4)	1622–2436 (1848, n = 12)
Testes number	63–69 (66, n = 2)	41–45 (43, n = 7)
Testes length (average)	26–28 (27, n = 2)	47–72 (53, n = 5)
Testes width (average)	20–22 (21, n = 2)	74–108 (92, n = 5)
Testicular field length	613–766 (689, n = 2)	1165–1862 (1491, n = 8)
Testicular field (percentage of body length)	33% (n = 2)	33–40% (36%, n = 8)
Vesicula seminalis length	86–112 (99, n = 2)	187–268 (234, n = 6)
Vesicula seminalis wide	11–16 (13.5, n = 2)	23–43 (31, n = 6)
Cirrus sac length	86–114 (100, n = 2)	184–283 (226, n = 7)
Cirrus sac width	30–49 (39.6, n = 2)	60–94 (76, n = 7)
Cirrus length	-	47–65 (55, n = 3)
Cirrus width	24–34 (29, n = 2)	37–60 (45.27, n = 3)
Ovary length	99–124 (111, n = 2)	174–342 (244, n = 8)
Ovary width	99–124 (111, n = 2)	115–306 (211, n = 8)
Ovary from its posterior end to posterior end of body	481–597 (539, n = 2)	778–1100 (911, n = 8)
Ovary from its posterior end of body end (percentage of body length)	25–26% (25.5%, n = 2)	21–23% (22%, n = 8)
Oviduct length	163–174 (169, n = 2)	183–336 (262, n = 7)
Oviducal seminal receptacle length	81–93 (87, n = 2)	174–254 (214, n = 7)
Oviducal seminal receptacle width	19–21 (20, n = 2)	34–54 (44, n = 7)
Oötype length	20–22 (21, n = 2)	42–66 (57, n = 6)
Oötype width	14–18 (16, n = 2)	33–75 (50, n = 6)
Oötype from posterior end of body	202–236 (219, n = 2)	330–443 (361, n = 6)
Female genital pore from sinistral body margin	28–31 (30, n = 2)	102–174 (143, n = 5)
Distance between male and female genital pores	145–208 (177, n = 2)	167–204 (190, n = 5)
Female genital pore from posterior end of body	409–548 (478, n = 2)	616–753 (664, n = 5)
Eggs length	18–21 (20, n = 20)	26–37 (31, n = 15)
Eggs width	13–16 (15, n = 20)	28–28 (24, n = 15)

## Data Availability

The new data are available at https://www.ncbi.nlm.nih.gov/genbank/, accessed on 15 March 2021.

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
