# Peer review of "First Report of Blood Fluke Pathogens with Potential Risk for Emerging Yellowtail Kingfish (Seriola lalandi) Aquaculture on the Chilean Coast, with Descriptions of Two New Species of Paradeontacylix (Aporocotylidae)"

_pathogens, 2021, doi:10.3390/pathogens10070849_

Round 1

Reviewer 1 Report

Sepúlveda and colleagues presents a parasitological study focused on blood flukes (Digenea) belonging to the genus Paradeontacylix of wild and farmed fish Seriola lalandi. Based on morphology, morphometry, and molecular analyses, authors propose two new species as Paradeontacylix humboldti n. sp. and Paradeontacylix olivai n. sp. This paper is a relevant and rigorous study of fish parasites which can cause economic loss to aquaculture production, moreover, it may also be of interest to helminthology specialists worldwide. I recommend publication of the manuscript, with only a few minor corrections. Please find more specific comments below.

ABSTRACT

Line 13 Please, rephrase as follows: Larval stage of these parasites are undetectable based on macroscopic observations in the definitive host, with the infection becoming evident when eggs accumulate in the branchial vessels.

Line 15-16 Please, rephrase as follows: Actually, there are nine known species of the genus Paradeontacylix and seven exclusively parasitize the Seriola spp. from several geographical areas.

INTRODUCTION

Line 88-90

Authors state that the fish death was due to the parasite. Did you perform an anatomopathological examination? If you did, please provide more detail in materials and methods and results. In addition, you should discuss your personal observation, in my opinion these data improve the value of this study.

Line 98-99

Can you provide more information of the origin of wild fish samples?

MATERIALS AND METHODS

As I commented above, please, provide more detail of anatomopathological examination

RESULTS

I recommend to provide parasitological indexes as mean abundance, mean intensity and prevalence and cite in materials and method Bush, A. O., Lafferty, K. D., Lotz, M., & Shostak, A. W. (1997). Parasitology meets ecology. The Journal of Parasitology, 83(4), 575–583. https://doi.org/10.2307/3284227.

3.3. Morphologic descriptions

I strongly recommend to configure a table that reports the measures effectuated (maximum width, marginal tegumental spine rows, posterior spine length, esophagus length, anterior caeca–intestine distance, posterior caeca–intestine distance, testicular area length, male genital pore–posterior end distance, ovary length, ovary width, oviducal seminal receptacle length, oviducal seminal receptacle width, oötype length, oötype width and female genital pore–posterior end distance) for both Paradeontacylix humboldti n. sp. and Paradeontacylix olivai n. sp. In my opinion a table provides clarity to the text and is easier to consult.

Figures 1-2-3 I strongly recommend to provide images with at least 300 dpi resolution.

Figure legends:

Figure 4 and 5

I recommend to write in bold the abbreviations in the legend of the above mentioned images in order to make faster their consultation

Author Response

Replay to Reviewer 1:

Sepúlveda and colleagues present a parasitological study focused on blood flukes (Digenea) belonging to the genus Paradeontacylix of wild and farmed fish Seriola lalandi. Based on morphology, morphometry, and molecular analyses, authors propose two new species as Paradeontacylix humboldti n. sp. and Paradeontacylix olivai n. sp. This paper is a relevant and rigorous study of fish parasites which can cause economic loss to aquaculture production, moreover, it may also be of interest to helminthology specialists worldwide. I recommend publication of the manuscript, with only a few minor corrections. Please find more specific comments below.

ABSTRACT

Line 13 Please, rephrase as follows: Larval stage of these parasites are undetectable based on macroscopic observations in the definitive host, with the infection becoming evident when eggs accumulate in the branchial vessels. R: Okay.

Line 15-16 Please, rephrase as follows: Actually, there are nine known species of the genus Paradeontacylix and seven exclusively parasitize the Seriola spp. from several geographical areas.R: Okay

INTRODUCTION

Line 88-90: Authors state that the fish death was due to the parasite. Did you perform an anatomopathological examination? If you did, please provide more detail in materials and methods and results. In addition, you should discuss your personal observation, in my opinion these data improve the value of this study.

R: Information was included in methods and discussion as suggested by the reviewer.

Line 98-99: Can you provide more information of the origin of wild fish samples? R: Yes. Please see line 99.

MATERIALS AND METHODS

As I commented above, please, provide more detail of anatomopathological examination. R: we have added information about appearance of specimen as follows: The fish was characterized by opened mouth and opercula, showing typical sign of suffocation. Please see lines 1.5-1.6.

RESULTS

I recommend to provide parasitological indexes as mean abundance, mean intensity and prevalence and cite in materials and method BushBush, A. O., Lafferty, K. D., Lotz, M., & Shostak, A. W. (1997). Parasitology meets ecology. The Journal of Parasitology83(4), 575–583. https://doi.org/10.2307/3284227. R: It was included. See lines 111-112.

R: this information was included for P. olivai n sp, but P. humboldti n sp was recorded only in farmed fish. Please see line 196.

3.3. Morphologic descriptions

I strongly recommend to configure a table that reports the measures effectuated (maximum width, marginal tegumental spine rows, posterior spine length, esophagus length, anterior caeca–intestine distance, posterior caeca–intestine distance, testicular area length, male genital pore–posterior end distance, ovary length, ovary width, oviducal seminal receptacle length, oviducal seminal receptacle width, oötype length, oötype width and female genital pore–posterior end distance) for both Paradeontacylix humboldti n. sp. and Paradeontacylix olivai n. sp. In my opinion a table provides clarity to the text and is easier to consult.

R:  We included Table 3 showing body measurements as suggested by reviewer.

Figures 1-2-3 I strongly recommend to provide images with at least 300 dpi resolution.

R: Okay.

Figure legends:

Figure 4 and 5: I recommend to write in bold the abbreviations in the legend of the above-mentioned images in order to make faster their consultation. R: Okay.

Reviewer 2 Report

I was honored to review the manuscript entitled “First report of blood fluke pathogens with potential risk factor emerging yellowtail kingfish (Seriola lalandi) aquaculture on the Chilean coast, with descriptions of two new species of Paradeontacylix (Aporocotylidae)” submitted to Pathogen. The study presents good quality and deals with first time report of two blood flukes based on morphology, molecular diagnostics and morphometry. However, there are some points that the authors should correct them and the manuscript needs some edits in English language and style.

There are some points to correct:

  • Please provide full author names in the front matter.
  • The authors should provide the manuscript according to journal guideline (Introduction, Results, Discussion, Material and Method, Conclusion).
  • It is more suitable to explain briefly about which methods that they used in this survey in the abstract.
  • It would be more better to add a sentence about the clinical signs of this disease in introduction.
  • The authors need to include PCR pictures, DNA sequences (supplement material) and add some data about the method of sequencing (illumina or sanger or….).
  • Page 4, Line 194, provide supplementary material at the back matter and describe the figure 1 there.
  • Prevalence rate of homboldti n.sp.?
  • Page 9, Line 437: Please clarify which terebellid polychaetes species are known as a intermediate host.
  • The discussion section needs improvement.
  • It would be better, the authors provide conclusion and abbreviation.
  • Please provide author contributions according to journal format.  
  • Please check all references and write them according to journal format. You can find some example blow.
  • Check reference number 7 line 500 (2016).
  • Reference 11: Italic
  • Reference 13:( 1934)
  • Ref 16, 17, 19, 20, 23, 27, 28, 29, 30, 37, 41, 48, 50, 56, 60.
  • Please add editors to reference 33.
  • The photographs were used in figure 1 had a poor quality. Please provide high quality pictures and phylogenetic tree.

There are some points for English language and style:

Line 2: Blood should be not bold.

Line 20-22: Please check the grammar and rewrite the sentence.

Line 232: Italic

Line 234: The species should be italic.

Line 309 & 311: Italic

Line 341-347: Check the grammar and rewrite the sentences.

Line 357: Check grammar

Line 368: has

Line 369: Using as well as instead of and (testes as well as fewer and smaller).

Line 389: are

Line 390: an important as a first step….

Line 391: Rewrite the sentence.

Line 458 & 459: were or was?

In conclusion I believe, in my opinion that the present manuscript can be accepted after major revision, for publication.    

Round 2

Reviewer 2 Report

I would like to thank the authors for correcting the manuscript however it needs more correction. Please provide full author names in the front matter (You did not correct this point in the present format) Line 48: By an mouth? Line 466: Eggs Line 468: Italic Author contribution is not according to guideline (Example: Conceptualization, F. A. S) Reference 5: Ass Fish Pathol.1992 (after name of journal you should use full spot). I recommended you before to correct them but unfortunately, several of still need correction. Please provide correct version. Please clarify, where is the Table 3 that you mentioned in line 145? Figure 1, 2 and 3 still need to improve the quality.

Author Response

Comments and Suggestions for Authors

 I would like to thank the authors for correcting the manuscript however it needs more correction. Please provide full author names in the front matter (You did not correct this point in the present format)

Line 48: By an mouth?

R: Corrected

Line 466: Eggs

R: Corrected (line 461)

Line 468: Italic Author contribution is not according to guideline (Example: Conceptualization, F. A. S) R: Corrected.

Reference 5: Ass Fish Pathol.1992 (after name of journal you should use full spot).

I recommended you before to correct them but unfortunately, several of still need correction. Please provide correct version.

R: Revised.

Please clarify, where is the Table 3 that you mentioned in line 145?

R: Now, we clary in the descriptions of the new species that body measurements are given in Table 3. See line 153, 154, and lines 222-223, respectively.

Figure 1, 2 and 3 still need to improve the quality

R: The quality pf figures was improved.